# High-Precision Optical Coherence Tomography Navigated Laser Retinopexy for Retinal Breaks

**DOI:** 10.3390/life13051145

**Published:** 2023-05-09

**Authors:** Simon Salzmann, Philip Wakili, Sami Al-Nawaiseh, Boris Považay, Christoph Meier, Christian Burri

**Affiliations:** 1Institute for Human Centered Engineering (HuCE)-OptoLab, Bern University of Applied Sciences, Quellgasse 21, 2501 Biel, Switzerland; 2Eye Clinic Sulzbach, Knappschaftsklinikum Saar, An der Klinik 10, 66280 Sulzbach, Germany; 3Department of Ophthalmology, University Hospital Muenster, Domagkstrasse 15, 48149 Muenster, Germany; 4Biomedical Photonics Group, Institute of Applied Physics (IAP), University of Bern, Sidlerstrasse 5, 3012 Bern, Switzerland

**Keywords:** OCT, guided laser, photocoagulation, LPC, retinal detachment prophylaxis, retinal tear

## Abstract

The prevalent cause of retinal detachment is a full-thickness retinal break and the ingress of fluid into the subretinal space. To prevent progression of the detachment, laser photocoagulation (LPC) lesions are placed around the break in clinical practice to seal the tissue. Unlike the usual application under indirect ophthalmoscopy, we developed a semi-automatic treatment planning software based on a sequence of optical coherence tomography (OCT) scans to perform navigated LPC treatment. The depth information allows demarcation of the border where the neurosensory retina is still attached to the retinal pigment epithelium (RPE), which is critical for prevention of detachment progression. To evaluate the method, artificially provoked retinal breaks were treated in seven ex-vivo porcine eyes. Treatment outcome was assessed by fundus photography and OCT imaging. The automatically applied lesions surrounding each detachment (4.4–39.6 mm^2^) could be identified as highly scattering coagulation regions in color fundus photography and OCT. Between the planned and applied pattern, a mean offset of 68 µm (SD ± 16.5 µm) and a mean lesion spacing error of 5 µm (SD ± 10 µm) was achieved. The results demonstrate the potential of navigated OCT-guided laser retinopexy to improve overall treatment accuracy, efficiency, and safety.

## 1. Introduction

Rhegmatogenous retinal detachment is the most common type of retinal detachment (RD) and is caused by a full-thickness retinal break. This complete opening of the neurosensory retina (NSR) allows the entry of liquefied vitreous into the subretinal space and the formation of subretinal fluid (SRF) as an intersecting layer in the break’s vicinity. The separation between the NSR and the retinal pigment epithelium (RPE) leads to a gradual impairment and successive death of the photoreceptors, due to interruption of the nutrient supply and the visual cycle. As a consequence, untreated RD can lead to temporal and further profound vision loss [1]. Retinal breaks with minimal adjacent RD can be treated prophylactically by unphysiologically connecting retinal and choroidal tissue via focal scar tissue formation in the vicinity of tears, or in extended rows to build a secondary ora serrata that stabilizes the tissue stack and improves retinal adhesion. Retinopexy was traditionally performed by cryotherapy that has been widely replaced by laser-therapy with the advent of the indirect ophthalmoscope and the simpler optical access. Other techniques that lead to long-term mechanical connection of the layers that can withstand high shear forces, such as high-frequency electrical welding have been investigated nevertheless, contact-free optical access is an outstanding advantage to control the position of lesions and ensure sterility [2]. The treatment aims to penetrate the choroidal-retinal interface, formed by the RPE, and induce ingrowth succeeded by scar formation and thus prevent the progression of the retinal separation that is caused by tensile and compressive stress induced by SRF penetrating through the retinal break and interaction with the vitreous [3]. Cryotherapy involves holding a cold metal probe against the outer wall of the eye around the retinal break until the ophthalmoscopically observed whitening of the retina is visible [4,5]. Laser retinopexy achieves the sealing by applying several rows of laser photocoagulation (LPC) lesions around the break. The local absorption of light energy at each lesion results in heat and causes protein denaturation, which is also ophthalmoscopically visible as whitened or blanched spots due to increased light scatter [6,7]. Cryotherapy is usually the preferred treatment in eyes with opaque media or small pupils, laser retinopexy remains the standard otherwise, as there is less collateral damage to the surrounding retina [3].

Laser retinopexy is conventionally performed via a slit lamp over a wide-field lens or a three-mirror lens. The technique requires experience and some manual dexterity on the part of the operator to be successful. Garoon et al., showed that 16% of eyes treated by experienced physicians required retreatment, while the number increased to 35.1% when assistants performed laser retinopexy [8]. Others reported this figure to be between 24% and 40% [9,10]. The increased rate of unsuccessful treatments could be due to missing spots, insufficient coagulation, or too large gaps between lesions in the manual procedure. To eliminate the difficulty of manually guiding the laser while constantly adjusting the imaging device, the use of a navigated retinal laser to treat retinal breaks as RD prophylaxis was first introduced by Gologorsky et al., in 2018 [11].

A major challenge during conventional treatment is the correct evaluation of the retinal break’s extent. The transition from RPE-attached to detached NSR is not visible ophthalmoscopically due to the lack of depth resolved information of the tens of micrometer thick fluid-filled gap that generates no significantly different accumulative reflective behavior. Therefore, it is up to the experience of the operator to accurately estimate the extent of the delamination region, as well as to predict the strength of required laser tissue interaction to foresee the interventions outcome and judge between irreversible damage due to further spread of the tear or extensive scarring. While the examination of the fundus is the most common method to detect retinal breaks and the formation of SRF, optical coherence tomography (OCT) can provide more reliable and substantiated information about the extent of the damage. Depth-resolved OCT B-scans separate the individual reflections formed by the two additional surfaces formed by the RPE-fluid and fluid-RPE interfaces and can therefore clearly reveal the retinal break. An extended scan that covers the complete potentially affected volume is able to delineate the delamination area, which is an eminently important factor for a successful treatment and to prevent break progression. Apart from detailed and reliable treatment planning, the minimally burdening and well-tolerated OCT scan also enables detailed post-treatment analysis for laser coagulation verification and reduction of the therapeutically induced collateral damage. With access to more sophisticated and finer adjustable treatment options the additional information provided by high-resolution demarcation of affected areas has the potential to improve the outcome of these surgical interventions.

Therefore, this study aimed to combine OCT imaging for reliable detection of retinal breaks, fundus imaging for planning and control of treatment position, and a navigated laser for optimal laser placement and application. The platform used for this purpose is a noncommercially available prototype that combines live fundus imaging, OCT, and a steerable treatment laser [12]. The goal was to develop and implement a novel method for navigated, OCT-guided treatment of retinal breaks.

## 2. Materials and Methods

### 2.1. Treatment and Monitoring System

The laser treatment system used in this study was the non-commercially available research device called Spectralis Centaurus (HuCE-optoLab, Bern University of Applied Sciences, Biel, Switzerland) [12]. This system is based on a diagnostic imaging system (Spectralis, Heidelberg Engineering, Heidelberg, Germany) and was extended by an LPC treatment laser (Merilas 532 shortpulse, Meridian Medical, Thun, Switzerland). The system allows acquisition of cross-sectional and volumetric spectral domain (SD) OCT images (B- and C-scan), plannable confocal scanning laser ophthalmoscope (cSLO) guided LPC treatment, and follow-up examinations. The diagnostic imaging part of the system is based on a SD-OCT at a wavelength of 880 nm and a spectral bandwidth of 73 nm. An additional infrared cSLO at 815 nm allows for live fundus imaging. The treatment laser radiates light at 532 nm and achieves a top-hat square beam profile of 90 × 90 µm^2^ on the ex-vivo porcine retina.

### 2.2. Treatment Planning

An in-house developed treatment software was used to operate the Spectralis Centaurus, providing interaction between the LPC laser and imaging hardware. To realize the elaborated retinal break treatment concept, an additional treatment planning software was developed and integrated into the system. For this purpose, the available XML data export of the image data management software (HEYEX, Heidelberg Engineering, Heidelberg, Germany) was used. The basic concept is shown schematically in Figure 1, whereby the elaborated procedure followed the subsequent steps: locate a retinal break, acquire cross-sectional SD-OCT scans to assess the extent of SRF, manually mark the outer SRF-border in SD-OCT-scans distributed around the break, calculate a treatment region based on these markings, and apply the LPC treatment accordingly. The software was implemented in C++, in a Qt environment (Qt 5.13.1, The Qt Company, Espoo, Finland) using a Microsoft Visual Compiler (MSVC2017, Microsoft, Washington, DC, USA) and CMake (CMake 3.16.0, Kitware, New York, NY, USA) for build automation.

The prerequisite for precise treatment planning is a volumetric SD-OCT scan that completely captures the retinal break and the adjacent areas of detachment. The image quality of the SD-OCT scans should be of sufficient quality, particularly the lateral positions of the detachment boundary between RPE and NSR must be discernible. As provided by the imaging platform, a C-scan or radial scan pattern can be used for volumetric data acquisition.

Laser retinopexy for prophylactic sealing retinal breaks involves applying multiple rows of LPC treatments around the break in the area of still attached NSR on the RPE. With its high lateral and axial resolution, SD-OCT is an excellent tool to find the outer boundary of the detached NSR. To ensure the treatment is performed exclusively in the intended area, the realized software allows to manually mark the detachment boundaries in the dataset (Figure 2). Only a few markings distributed around the break are sufficient for subsequent determination of the treatment region.

After placing at least four detachment markers, the treatment region can be calculated. To ensure a treatment solely outside of the detachment boundary, the treatment area should enclose all marked points. This also prevents the result from being affected by incorrectly placed markers that are too close to or within the retinal break. To minimize focal stress on the tissue and mitigate retinal damage, the smallest possible elliptical treatment area is fitted. The calculation is performed by finding the minimum-area enclosing ellipse around a given set of points. The applied method is based on the Khachiyan algorithm [13] used for calculating the minimum-volume enclosing ellipsoid (MVEE), adapted from the implementation of Jambawalikar and Kumar [14], approximated as
(1)$$MVEE\left(P\right)={\left(P-c\right)}^{T}A\left(P-c\right)\leq 1$$
with
(2)$$A=\frac{1}{d}{\left(PUP^{T}-Pu{\left(Pu\right)}^{T}\right)}^{-1}$$
and
(3)c=Pu
where *u* is the decision variable, *U* a matrix with diagonal entries of *u*, *P* the initial points, *c* the center, and *d* the dimension.

Based on the optimized ellipse around the marked points, a treatment area with user-definable width and distance to the markers is calculated. The computed region is indicated on the fundus image and displayed in the affected SD-OCT B-scan, as shown in Figure 2.

Treatment points were placed to achieve an even and user-definable distribution (point and row spacing) in the previously identified treatment region. The resulting treatment points are displayed to the user in the fundus image (Figure 2a) and the corresponding coordinates exported for treatment execution. 

### 2.3. Explant Preparation and Treatment

Retinal breaks were provoked in a total of seven enucleated porcine eyes from a local slaughterhouse to evaluate the elaborated treatment method. Using a hypodermic needle inserted into the ora serrata and a saline-filled syringe, a retinal break was induced close to the optical axis in each explant. The RD around the break was provoked by slightly changing the syringe pressure with the needle tip placed on the incision. Intraocular pressure (IOP) was measured and externally compensated to maintain a pressure of 18 ± 3 mmHg during the procedure and the eyes were periodically moistened with saline solution. 

Three rows of contiguous treatment points were placed around each RD with a radial distance (row to row) and tangential spacing (point to point) of 200 µm and 300 µm, respectively. LPC lesions were applied with a power of 200 mW for 200 ms. Before and after the treatment, SD-OCT C-scans and cSLO images were acquired using the standard 30° lens. Color fundus photography (CFP) was assessed using a fundus camera (Fundus Module 300, Haag-Streit, Köniz, Switzerland). Post-processing and evaluation were conducted using the Fiji image processing package distribution of ImageJ [15].

## 3. Results

Each of the seven enucleated porcine eyes was successfully prepared with an artificially induced retinal break of varying extent (4.4–39.6 mm^2^) and subsequently treated with 99 to 227 lesions. The elliptical treatment regions ranged in size from 4.5 mm to 8.7 mm along the major axis. Treatment time per break varied from 2:11 min to 4:19 min, with an average treatment time per lesion of 1.2 s. The total time per eye, including examination, planning, and treatment, averaged 8:42 min.

After treatment, CFP and cSLO showed the typical tissue whitening caused by LPC lesions around the retinal break (Figure 3). The SD-OCT B-scans showed hyperreflectivity of the retina at the laser application sites, indicating successful treatment. In most eyes, the laser treatment was entirely applied in regions of still attached NSR, the goal of successful laser retinopexy to seal the break. Two exemplary results are shown in Figure 4 and Figure 5.

Comparing the planned treatment patterns and the cSLO images after treatment for eight laser lesions (evenly distributed around each break), a mean offset of 68 µm was found (median: 67 µm, standard deviation (σ_n−1_): ±16.5 µm, number of samples: 56). Measuring the tangential distances between two adjacent lesions at the same eight sites around each break resulted in an average spacing error of 5 µm (median: 5 µm, standard deviation (σ_n−1_): ±10 µm, number of samples: 56).

## 4. Discussion

While the use of navigated laser retinopexy based on live fundus imaging using a SLO was reported previously [11,16], no known implementation has used OCT imaging for treatment planning and subsequent navigated LPC (literature research: Appendix A). Based on studies demonstrating the beneficial impact of OCT diagnosis on retinal breaks and RD [17,18], planning guided by OCT should allow for more adapted and effective treatment. Experience and a certain degree of manual dexterity are required to perform conventional laser retinopexy under indirect ophthalmoscopy. Several reports found substantial differences in retreatment rates between experienced physicians (16%) and physicians in training (24% to 35.1%) [8,9,10].

The method proposed in this study has shown to perform automated laser application according to a predetermined treatment plan at high accuracy (mean offset of 68 µm, mean tangential spacing error of 5 µm between lesions), which could therefore support less experienced users, but also reduces patient discomfort and treatment time and allow further improvements in treatment planning.

No planned treatment point remained untreated or was visibly misplaced. However, in some samples, torn tissue from the artificially induced retinal breaks, which obstructed parts of the treatment area, hindered proper SD-OCT acquisition and presumably laser application to the underlying retina. In a clinical situation, motion of tissue between initial measurement, planning and treatment typically is significantly lower than in this investigation. Nonetheless such changes might be addressed by an additional software module that indicates or even reacts on changes by interrupting or altering the exposure.

To ensure that each treatment point receives the correct energy density, the treatment laser focal point must remain on the retina. Conventional treatment approaches via a slit lamp or indirect laser ophthalmoscope necessitate constant manual alignment of the focus requiring manual dexterity and experience. In the presented method this task is carried out by the alignment system in all three dimensions, thereby significantly facilitating the treatment process.

The presented experiments were performed on ex-vivo porcine eyes within a few hours after enucleation allowing for ideal fundus visibility and OCT image quality. However, in clinical application fundus visibility and OCT image quality can be impaired by several factors such as the presence of cataracts, small pupils, gas tamponade, eye moisture and the location of the target area of the treatment in the periphery of the eye. In addition, acquiring high-quality and large-area OCT images can be difficult for both the clinician and the patient and depends also on the device used. In order to plan the treatment with the proposed method, proper OCT scans covering the affected area including adjacent SRF are a critical factor. Since the applied LPCs need to surround the retinal break, the fundus image must cover an even larger area. Further validation studies are required to determine how far the presented benefits of the proposed treatment procedure also apply in non-optimal settings.

To artificially provoke retinal breaks with parallel cSLO imaging, the incisions were performed close to the optical axis and therefore no treatment planning and subsequent laser retinopexy was performed in the periphery of the porcine eyes. Further investigations are required to show the feasibility of the presented approach in conjunction with the Spectralis Centaurus for peripheral or even anterior retinal breaks. In this regard, Choudhry et al., previously used a similar OCT system with an attached 30° lens to acquire OCT scans covering a total field of view (FOV) of 200°, which reached across the ora serrata [19]. This required an optimized medication for pupil dilation, an experienced surgeon, and the patient’s cooperation. Another approach was reported by Brodie et al., and involved a Goldmann three-mirror contact lens for OCT acquisition of anterior retinal breaks, which may represent a technique for laser application as well [20]. To expand its use as a broadly applicable clinical tool, the treatment of peripheral or even anterior located retinal breaks must be investigated in future studies.

Future work on the detection side aims to automate the RD labeling of OCT data by image processing and segmenting B-scans. This permits to process larger and more precise scans where the quality of the OCT acquisition might be of even greater importance. However, if such segmentation proved successful and robust, this would allow for considerably faster, facilitated, and highly automated treatment planning and verification. Additionally, the device’s ability to re-investigate segmented areas of detachment during long-term follow-up would help provide a better understanding of indications and treatment parameters, such as the shape and size of the detachment on the risk of later re-detachment or the lateral exposure density and required radiant exposure per treatment spot. It is expected that this precise tool helps better adopt to the individual needs of patients, e.g., by altering the number of lesions around the break, the radial spacing between rows, the tangential spacing between lesions or the exposure in the vicinity of retinal vasculature or even nerve fibers. The fact that OCT is significantly more sensitive allows to investigate and treat retinal breaks with higher accuracy and possibly with a less invasive approach.

## 5. Conclusions

To our knowledge, this is the first report of an automated retinopexy system with high-resolution treatment planning and precise laser spot placement. SD-OCT as imaging modality to plan laser retinopexy shows significant advantages in finding the optimal treatment region compared to fundus observation and is the key to successful treatment. While capturing the entire retinal break including adjacent RD is crucial, the number of acquired B-scans is of minor importance. The investigation of fundus photographs after LPC application demonstrated typical tissue whitening at intended target points. The treatment spots were evenly distributed surrounding the breaks, according to the user-defined parameters for radial and tangential points spacing. SD-OCT revealed the ruptured retina surrounding the RD, indicating reliable sealing of the break. The successful ex-vivo experiments demonstrated the potential of the novel method and indicated improved applicability and accuracy compared to conventional laser retinopexy. Nonetheless, the system platform stems from clinical imaging and has already proven that in-vivo human application of the individual components can be performed. As shown in this study, the coalescence of high-precision metrology and therapy also enables pre- and post-examination of the intervention, in addition to planning and execution. Future studies are required to determine and validate the optimal treatment parameters, investigate the treatment of far-peripheral retinal breaks, and evaluate the impact of fundus visibility impairment.

Unlike conventional devices, the Spectralis Centaurus offers the advantage of keeping the treatment laser constantly in focus and laterally positioned even on the non-retained retina by utilizing an eye tracker that was not required in the current study to maintain the performance also in a clinical setting. Utilizing the full feature set of this sophisticated metrology device in conjunction with navigated laser coagulation has the potential to significantly improve the outcome for treatment of retinal detachment.

## Figures and Tables

**Figure 1 life-13-01145-f001:**
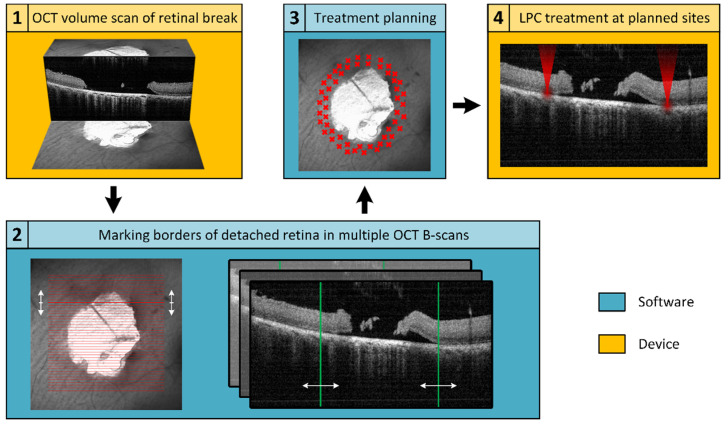
Schematic representation of the workflow for optical coherence tomography (OCT)-based semi-automatic treatment of retinal breaks. The first step (1) is to acquire an OCT volume scan (C-scan or radial scan) covering the retinal break and adjacent detached areas. Subsequently, (2) the OCT data are imported into a treatment planning software, where the user encircles the retinal detachment (RD) (still attached neurosensory retina (NSR) to the retinal pigment epithelium (RPE)) by investigating the OCT B-scans. Based on the demarcation, an optimal treatment area is calculated (3) and finally applied (4).

**Figure 2 life-13-01145-f002:**
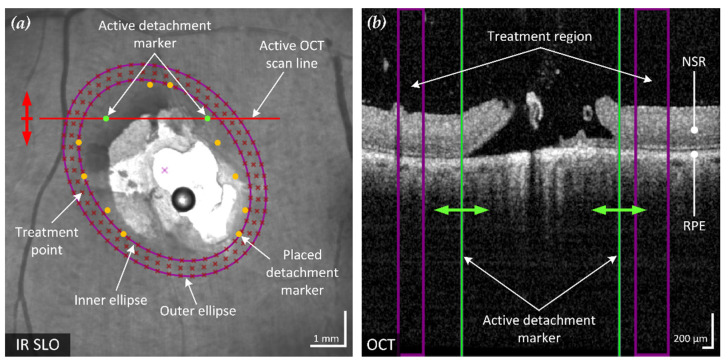
Example of the proposed method of an optimized elliptical treatment region and uniformly placed treatment points around a retinal break. The position of the active spectral domain (SD)-OCT scan (red line) in the confocal scanning laser ophthalmoscope (cSLO) fundus image (**a**) is movable according to the acquired B-scans (red arrow) and corresponds to the SD-OCT B-scan shown (**b**). The two green retinal detachment marker lines in the B-scan are moved by the user (green arrows) to mark the visible outer boundary of the detached NSR from the RPE. The six pairs of markers in the fundus image represent the marked detachment boundary of the annotated B-scans. Based on these markings, an enclosing ellipse with minimum area is fitted to optimally surround the break and adjacent RD (inner ellipse). Together with a similar but enlarged second ellipse (outer ellipse), the treatment region is defined. This region is also visualized in each B-scan (purple lines). In the example shown, three rows of evenly spaced treatment points have been placed within the treatment region.

**Figure 3 life-13-01145-f003:**
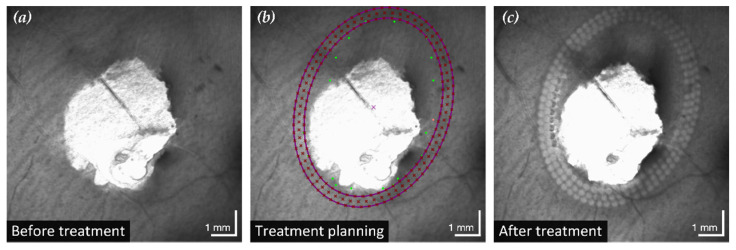
Example of initial situation (**a**), treatment planning (**b**) and outcome (**c**). While the cSLO image showed only the full-thickness retinal defect, SD-OCT based treatment planning revealed a large area of detached NSR in the upper quarter of the image. Based on the precisely determined detachment boundary, a minimum elliptical treatment area (violet ellipses) is calculated. Uniformly distributed treatment points (violet crosses) are subsequently positioned in this area and the LPC is applied accordingly.

**Figure 4 life-13-01145-f004:**
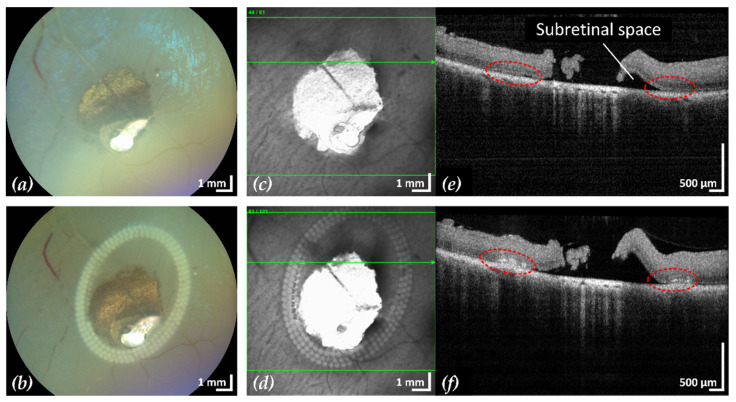
Example 1: treatment outcome in ex-vivo porcine eye with artificially induced retinal break (area of retinal break and adjacent RD: 25.2 mm^2^). Fundus photographs before (**a**) and after (**b**) treatment, infrared scanning laser ophthalmoscope images before (**c**) and after (**d**) treatment with the corresponding SD-OCT B-scans (**e**,**f**). A total of 200 lesions were applied in three rows with a radial and tangential (point to point) distance of 200 µm and 300 µm, respectively. The LPC treatment time was 4:19 min. The effect of LPC treatment is visible in (**b**,**d**) as spots of whitened tissue and in (**f**) as ruptures in the retina at the treatment sites (red).

**Figure 5 life-13-01145-f005:**
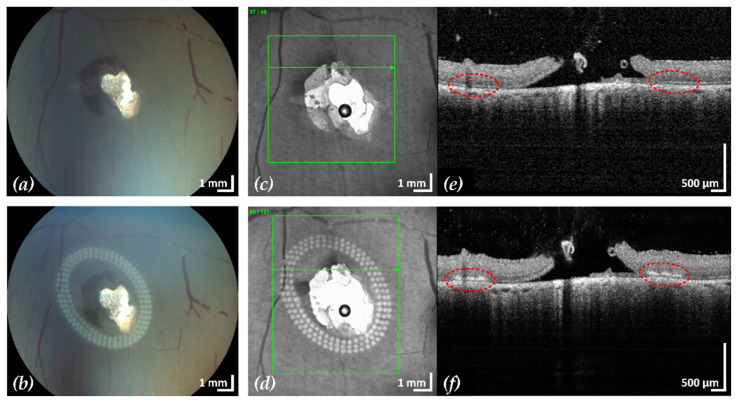
Example 2: treatment outcome in ex-vivo porcine eye with artificially induced retinal break (area of retinal break and adjacent RD: 15.4 mm^2^). Fundus photographs before (**a**) and after (**b**) treatment, infrared scanning laser ophthalmoscope images before (**c**) and after (**d**) treatment with the corresponding SD-OCT B-scans (**e**,**f**). A total of 153 lesions were applied in three rows with a radial and tangential (point to point) distance of 200 µm and 300 µm, respectively. The LPC treatment time was 2:11 min. The effect of LPC treatment is visible in (**b**,**d**) as spots of whitened tissue and in (**f**) as ruptures in the retina at the treatment sites (red).

## Data Availability

The data presented in this study are available on request from the corresponding author.

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
