# Peer review of "High-Precision Optical Coherence Tomography Navigated Laser Retinopexy for Retinal Breaks"

_life, 2023, doi:10.3390/life13051145_

Round 1

Reviewer 1 Report

This manuscript introduced an OCT based method for surgery planning in treatment of retinal detachment with laser photocoagulation. The manuscript was clear written with well organized results. But the work seems incremental. A control group or validation is needed for highlighting this work.

The manuscript was clearly written, and very readable.

Author Response

Thank you for the time you invested in reviewing our article and for the valuable feedback. Below is our response to your comment:

Reviewer Comment: The work seems incremental. A control group or validation is needed for highlighting this work.

Response: You are indeed correct about the incremental nature of the work, and further validation will be needed to substantiate this work and the idea behind. However, this was a first experiment and therefore can be considered more as a feasibility study, while further, more extensive experiments are planned for the near future.

Reviewer 2 Report

The article describes a successful experiment on the use of optical coherence tomography for laser navigation at retinopexy for retinal breaks. To the reviewer's knowledge, this is the first report of an automated retinopexy system with high-resolution treatment planning and precise laser spot placement. The author's technique has been demonstrated in an experiment with artificially provoked retinal breaks in seven ex-vivo porcine eyes and has shown good results. I agree, that this combine technique looks to perform automated laser application according to a predetermined treatment plan at high accuracy (mean offset of 68 μm, mean tangential spacing error of 5 μm between lesions), which could therefore support less experienced staff in particular, and also reduces treatment time and allow further improvements in treatment planning. The article is well written. There are no semantic comments on the article, therefore, it can be recommended to accept for publication in its current form. The only remark is: this article is not very similar to a scientific study. Rather it looks like a "case report". No any variants, different parameters etc. were studied in the “study”. Nevertheless, the article may be of interest to many readers as an interesting find.

Author Response

Thank you for the time you invested in reviewing our article and for your valuable feedback. Below is our response to your comment:

Reviewer Comment: This article is not very similar to a scientific study. Rather it looks like a "case report". No any variants, different parameters etc. were studied in the “study”.

Response: Regarding your comment on the publication format, we understand that no different parameters, variants, etc. were studied and therefore a "case report" might be an option. However, since only ex-vivo porcine eyes from slaughtered pigs were used and a "case report" almost exclusively describes cases from human patients, we believe that an article is the most appropriate format for this work. This was a first experiment for a novel treatment method and can therefore be considered more as a feasibility study, while further, more extensive experiments are planned for the future to validate and investigate for instance treatment parameters.

Reviewer 3 Report

The paper is written absolutely properly and does not need any improvements. The manuscript under consideration is a rare case of such a paper. No corrections or addenda are necessary.

Nice paper. Can be accepted as it is.

Author Response

Thank you very much for the time you invested in reviewing our article and for the highly positive feedback.